# Laboratory disruption of scaled astrophysical outflows by a misaligned magnetic field

G. Revet[1,2,3], B. Khiar[4,5], E. Filippov [1,6], C. Argiroffi[7,8], J. Béard [9], R. Bonito[8], M. Cerchez[10], S. N. Chen[1,11], T. Gangolf [2,10], D. P. Higginson[2,12], A. Mignone[13], B. Olmi [8,14], M. Ouillé [2], S. N. Ryazantsev[6,15], I. Yu. Skobelev[6,15], M. I. Safronova[1], M. Starodubtsev [1], T. Vinci [2], O. Willi[10], S. Pikuz [6,15], S. Orlando [8], A. Ciardi [4✉] & J. Fuchs [1,2✉]

The shaping of astrophysical outflows into bright, dense, and collimated jets due to magnetic pressure is here investigated using laboratory experiments. Here we look at the impact on jet collimation of a misalignment between the outflow, as it stems from the source, and the magnetic field. For small misalignments, a magnetic nozzle forms and redirects the outflow in a collimated jet. For growing misalignments, this nozzle becomes increasingly asymmetric, disrupting jet formation. Our results thus suggest outflow/magnetic field misalignment to be a plausible key process regulating jet collimation in a variety of objects from our Sun's outflows to extragalatic jets. Furthermore, they provide a possible interpretation for the observed structuring of astrophysical jets. Jet modulation could be interpreted as the signature of changes over time in the outflow/ambient field angle, and the change in the direction of the jet could be the signature of changes in the direction of the ambient field.

[1] Institute of Applied Physics RAS, Nizhny Novgorod, Russia. [2] LULI, CNRS, CEA, Sorbonne Université, École Polytechnique, Institut Polytechnique de Paris, Palaiseau, France. [3] Centre Laser Intenses et Applications, Université de Bordeaux-CNRS-CEA, Talence, France. [4] Sorbonne Université, Observatoire de Paris, PSL Research University, LERMA, Paris, France. [5] Flash Center for Computational Science, University of Chicago, Chicago, USA. [6] Joint Institute for High Temperatures RAS, Moscow, Russia. [7] Dipartimento di Fisica e Chimica, Universitá di Palermo, Palermo, Italy. [8] INAF-Osservatorio Astronomico di Palermo, Palermo, Italy. [9] LNCMI, UPR 3228, CNRS-UGA-UPS-INSA, Toulouse, France. [10] Institut für Laser und Plasmaphysik, Heinrich Heine Universität Düsseldorf, Düsseldorf, Germany. [11] ELI-NP, Horia Hulubei National Institute for Physics and Nuclear Engineering, Bucharest-Magurele, Romania. [12] Lawrence Livermore National Laboratory, Livermore, CA, USA. [13] Dip. di Fisica, Universiá di Torino, Torino, Italy. [14] INAF-Osservatorio Astrofisico di Arcetri, Firenze, Italy. [15] National Research Nuclear University 'MEPhl', Moscow, Russia. ✉email: andrea.ciardi@obspm.fr; julien.fuchs@polytechnique.fr

Outflows of matter are general features stemming from systems powered by: compact objects as varied as black holes, active galactic nuclei (AGNs), pulsar wind nebulae (PWNe); accreting objects as Young Stellar Objects (YSO); mature stars as our Sun, in the form of coronal outflows. In all these objects, varied morphologies are observed for the outflows, from very high aspect ratio, collimated jets, to short ones that are either clearly fragmented or are just observed not be able to sustain a high density over a long range. The mechanisms underlying these varied morphologies are however still unclear. What we will present here, and discuss in the light of observations made onto a variety of astrophysical objects, is a possible scenario where the relative orientation between the outflow and the large-scale ambient magnetic field surrounding the object can play a major role orienting the dynamics of the outflow from a collimated one to a stunted, fragmented one. Such a scenario was already evoked to explain the difference between confined and fragmented solar coronal outflows[1]. Here, we support it using laboratory experiments where we systematically vary the orientation between an outflow and an ambient magnetic field, and discuss its applicability to a large variety of astrophysical objects.

Clarifying the question of the dynamics leading to varied outflow morphologies is not limited to answering that sole question, but has also implications in helping understand the global dynamics of the source objects, since the outflows generation is intrinsically connected to the global dynamics of the objects. In YSO, for instance, the understanding of the outflow dynamics is crucial in acquiring a complete picture of the first stages of star formation. Indeed, it is only through the removal of angular momentum from the system, as provided by the outflow[2–4], that matter can be accreted on the star[5]. YSO jets are supersonically ejected into the ambient medium and often show a well collimated chain of knots detected in several bands, e.g., optical and X-ray bands[6]. YSO jets are detected during the early stages of evolution (class 0 and class I) and in Classical T Tauri Stars, when accretion of material onto the central object is still at work, while are not observed in more evolved stages when the accretion process is no longer active. YSOs outflow genesis is widely accepted, and originates from magneto-centrifugally accelerated disk or stellar winds[4,5,7]; the collimation process leading in some cases to remarkably narrow and stable jets[8] is however more controversial.

Bow-shock PWNe are another example of astrophysical sources where collimated jets can be observed. These nebulae are produced by fast moving pulsars (with velocities from hundreds to thousands of km s$^{-1}$) escaped from their parent supernova remnants[9,10]. Due to the supersonic motion of the pulsar, bow-shock PWNe show a cometary-like morphology with the pulsar located at the bright head of the nebula, and the long tail extending in the direction opposite to the pulsar motion (eventually for few pc[10]). Puzzling bright X-ray jets, largely misaligned with the pulsar direction of motion, have been observed in some bow-shock PWNe in the last years[11–15]. Their formation was recently clarified as the result of particles escaping the bow shock at magnetic reconnection locations with the interstellar magnetic field, developing at the magnetopause layer depending on the mutual inclination between the internal and outer fields[16]. Once escaped, particles then illuminate the structure of the ambient magnetic field[14,17,18]. However, the directionality of the jets, as well as the distortions that can be observed to affect some of them at certain distances from the pulsar, still remain to be clarified.

Several scenarios have been evoked to explain outflow collimation. As mentioned above, for solar coronal outflows, Petralia et al.[1] tested in simulations that the alignment of the flow with the local magnetic field lines plays a crucial role in the outflow morphology. Flows directed along the magnetic field lines were observed to be confined, while those having a slight misalignment with the magnetic field become fragmented, in agreement with solar observations. Also in the case of bow-shock PWNe there is some evidence for a direct impact of the geometry of the ambient magnetic field on the morphology of the system[16].

In YSO, the situation is more complex than for solar outflows, due to the varying magnetic field geometry as the flow propagates away from the star. At the launching stage, the flow is collimated by a toroidal magnetic field (magneto-hydrodynamic self-collimation)[19,20]; however, a dominant toroidal field, i.e., wound-up around the outflow, can potentially drive the jet unstable, as shown, for instance, in numerical simulations of YSO[21], and scaled laboratory experiments[22]. The collimation by a dominant poloidal magnetic field component, i.e., aligned with the outflow, through the pressure exerted by the magnetic field surrounding the flow, has been evoked as another plausible scenario[23–26]. This scenario was recently supported by laboratory experiments we performed[27,28], in which we showed that outflows having their axis aligned with that of the magnetic field result in long-range, stable and dense jets. However, YSO outflows are not expected to be necessarily aligned with the larger scale ~50 AU, non-local magnetic field surrounding the system[29]. Several observations of YSO have reported, at different length scales, the correlation between the axis of the outflows and that of the surrounding magnetic field[30–33]. Some studies[30,34–36] support the idea of randomly alignment, while a recent study[33] supports the idea of preferential alignment of the outflow with the magnetic field. Anyhow, when filtering the observations by looking at the degree of collimation of the outflows, both Strom et al.[37] and Ménard and Duchêne[36] highlight the preferential alignment of well collimated, bright, long-range jets with the magnetic field, while weaker or wider jets oppositely present a preferential misalignment.

We report here results related to a series of experiments that we have performed, to investigate the effect of a misalignment between an outflow and a poloidal magnetic field. Our findings support the idea that the alignment of the flow with the magnetic field plays a crucial role in allowing stable propagation of the flow. Note that this does not preclude a possible role played by a toroidal field, but it shows that a poloidal-only field influences strongly the outflow dynamics and morphology. We show that the experimental outflow scales well with YSO, as well as solar outflows. We also discuss the applicability of our findings to the morphology of other astrophysical objects that do not scale directly to the laboratory plasma.

## Results

In the laboratory experiment, a wide angle expanding plasma outflow, generated by ablating plasma from a solid by a high-power laser, is interacting with a large-scale magnetic field, homogeneous and permanent at the scales of the experiment, having a variable orientation with respect to the outflow main axis. As detailed in the Methods section and Table 1, such setup is shown to be scalable to a YSO wide angle outflow interacting with an ambient magnetic field within the 10 to 50 AU distance-from-the-source region of its expansion (see also refs. [27,28]), as well as to solar outflows[1]. As it will be discussed in the Methods section, the same scalability can not be applied in general to the case of PWNe, mainly due to the lack of knowledge of many of the parameters given for Solar outflows and YSO in Table 1.

As illustrated in Fig. 1A, by inclining the laser-irradiated target, we are able to vary the angle $\alpha$ of the magnetic field with respect to the main plasma flow direction. We demonstrate that (1) outflows tend to align over large scales with the direction of the magnetic field, even for an initial large misalignment of their axes,

**Table 1 Comparison and scalability between the laboratory, a YSO, and a coronal solar outflow.**

|  | Laboratory | YSO jet | Sun's coronal outflow |
|---|---|---|---|
| $B$-field [G] | $2 \times 10^5$ | $2.4 \times 10^{-2}$ | $30 - 3$ |
| Material | $CF_2$ (Teflon) | H | H |
| Atomic number | 16.7 | 1.28 | 1.29 |
| Spatial (Radial) scale [cm] | $1 \times 10^{-1}$ | $4.5 \times 10^{13}$ (3 AU) | $2 \times 10^8$ |
| Charge state | 8 | $2 \times 10^{-2}$ | 1 |
| Electron density [cm$^{-3}$] | $2 \times 10^{19}$ | $6.5 \times 10^4$ | $3 \times 10^{10}$ |
| Density [gcm$^{-3}$] | $7 \times 10^{-5}$ | $7 \times 10^{-18}$ | $6.4 \times 10^{-14}$ |
| Te [eV] | 300 | 3 | 3.4 |
| Flow velocity [kms$^{-1}$] | 550 (100-1000) | 250 (100-400) | 200 |
| $\beta_{dyn}$ | 133 | 191 | $0.7 - 7$ |
| Mach number | 3 | 13 | 1.5 |
| Alfvenic mach number | 3 | 8 | $0.5 - 1.4$ |
| Magnetic Reynolds number | $3 \times 10^3$ | $4 \times 10^{17}$ | $3.5 \times 10^{12}$ |
| Reynolds number | $5 \times 10^4$ | $2 \times 10^1$ | $1.7 \times 10^3$ |
| Peclet number | 2 | $2 \times 10^4$ | 31 |
| Euler number | 9 | 17 | 2 |
| Alfven number | $6 \times 10^{-4}$ | $20 \times 10^{-4}$ | $40 \times 10^{-4} - 4 \times 10^{-4}$ |

The YSO density, charge state, temperature and flow velocity are extracted from Maurri et al. and Ainsworth et al.[54,55], and correspond to the parameters of the DG Tau A object and its associated HH 158 jet in its launching region, i.e., just after a distance of 10 AU from the outflow source. The YSO jet spatial scale corresponds to the radius of the source at the outflow launching region, which was measured in detail in the HN Tau object[56] as 3 AU. The value for the magnetic field in the YSO outflow corresponds to that required to collimate the jet to its observed radius (see Supplementary Note 1). For the coronal solar outflow, the values are derived from the study of Petralia et al.[1]. The values of the magnetic field stated for that object correspond to the range existing between the foot (30 G) and the top (3 G) of a coronal loop. The value of the laboratory magnetic field (20 T) is chosen such that the laboratory plasma Alfven number is a best compromise match between the Alfven numbers of the YSO and of the solar corona outflows. The indicated velocity ranges correspond to the minimum and maximum speed within the flow; in the laboratory case it corresponds to the ballistic behavior of the expansion[64]. In the astrophysical cases, the material composition (indicated with H) consists in fully ionized hydrogen plus a mixture of heavier elements with abundances of 0.5 compared to the solar values[65]. N.B.: due to the very high Magnetic Reynolds number in all cases, and the associated very strong advection of the magnetic field lines as explained in the main text, the magnetic field is not expected to be presented in the core of the outflow; hence the Reynolds and Peclet numbers do not take into account any ion or electron magnetization correction. N.B. 2: the values in bold are measured or observed; the values in light are calculated or inferred.

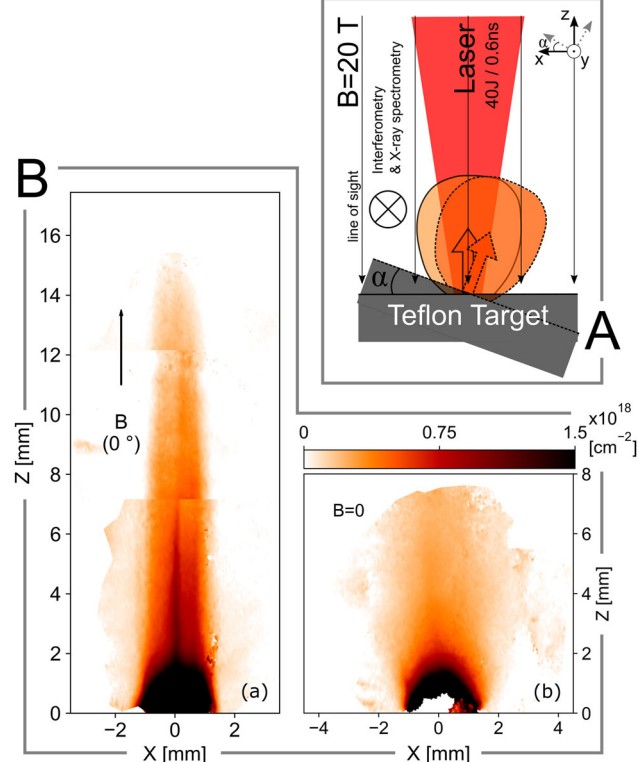

**Fig. 1 Schematic of experimental set-up and observed plasma expansion, with and without an aligned magnetic field. A**: Sketch of the experimental setup. The target is embedded in a large-scale 20 T magnetic field, and is heated by a $I_{max} = 1.6 \times 10^{13}$ W cm$^{-2}$, 0.6 ns duration laser. By tilting the target, it is possible to vary the angle $\alpha$ between the main plasma outflow direction and that of the magnetic field. The plasma is optically probed along the y axis. **B**: **a** Laboratory maps of the electron density integrated along the probe line of sight, in [cm$^{-2}$], retrieved via interferometric measurement (see "Methods" section), at 28 ns after the start of the plasma expansion. The vertical black arrows indicate the magnetic field direction; here aligned with the main axis of plasma expansion (i.e., along z; $z = 0$ being the target surface). **b** Same without any magnetic field applied.

and that (2) narrow collimation (i.e., the capability for the flow to keep a high density over large distance) is possible only for a small initial misalignment ($\lesssim 20-30°$). The latter is due to the fact that the generation of a diamagnetic cavity is only possible for a small misalignment. That cavity results from the plasma/magnetic field interaction[26]. Having shocked edges, the cavity forms an effective magnetic nozzle, which redirects the flow into a narrow, long range and high density jet[27,28]. These findings are corroborated by three-dimensional magneto-hydrodynamic (MHD) numerical simulations performed in laboratory conditions, and which will be detailed below.

The perfectly plasma/magnetic field aligned case, $\alpha = 0$, represents the ideal case for the collimation of the outflow by the poloidal magnetic field. The details of the collimation mechanism through the formation of an effective magnetic nozzle have been discussed in refs. [26–28] (the reader should refer to such references to get physical insights about the collimation mechanism). As illustrated by the Fig. 1Ba, b this magnetic nozzle allows the formation of a jet collimated over long spatial and temporal ranges, in contrast with the no-external-magnetic-field expansion, where a much faster density decrease along the outflow expansion can be observed.

By increasing the angle $\alpha$, we here after investigate the effect of a misalignment of the magnetic field on that collimated jet generation. The maps of Fig. 2a, b, c, for angles of 10, 20, and 45 degrees, respectively, present the electron density integrated along the probe line of sight (similarly as shown in Fig. 1Ba for the aligned case). In these maps, a clear curvature of the expanding plasma motion is observed and two stages are visible. Firstly, close to the target surface, the hot and dense plasma expands as

expected, i.e., perpendicularly to the target surface, pushing out the magnetic field lines. Later on, farther from the target surface, the plasma flow is observed to be redirected along the magnetic field, the funnelling tending to follow the original magnetic field axis. The dashed white lines superimposed on the electron density maps help guide the eyes on that redirection as they follow the center of mass of the plasma flow. Hereafter the axis following such line will be referred to as $z_{c.m.}$. This plasma redirection is also corroborated by looking at the X-rays emission from the plasma, see Fig. 2d, e, f. Note that the X-ray data also demonstrate that, in the presence of the magnetic field, the plasma exhibits higher temperature than that without magnetic field, irrespective of the misalignment between the flow and the magnetic field (see Supplementary Note 2 for details on the analysis). This latest experimental evidence highlight that, whatever the asymmetry of the plasma expansion is, a global heating of the plasma (through super-Alfvenic shocks[26]) occurs.

This redirection behavior is observed to be true for small misalignments, with jets forming and being oriented along the initial magnetic field orientation up to tens of millimeters away from the target (see 10° and 20° cases at 36 ns in Fig. 2a and b, respectively). However, for large misalignment cases, this seems to hold only for short times (see 45° case at 16 ns in Fig. 2c). Indeed, at later times, the outflow is unable to expand to large

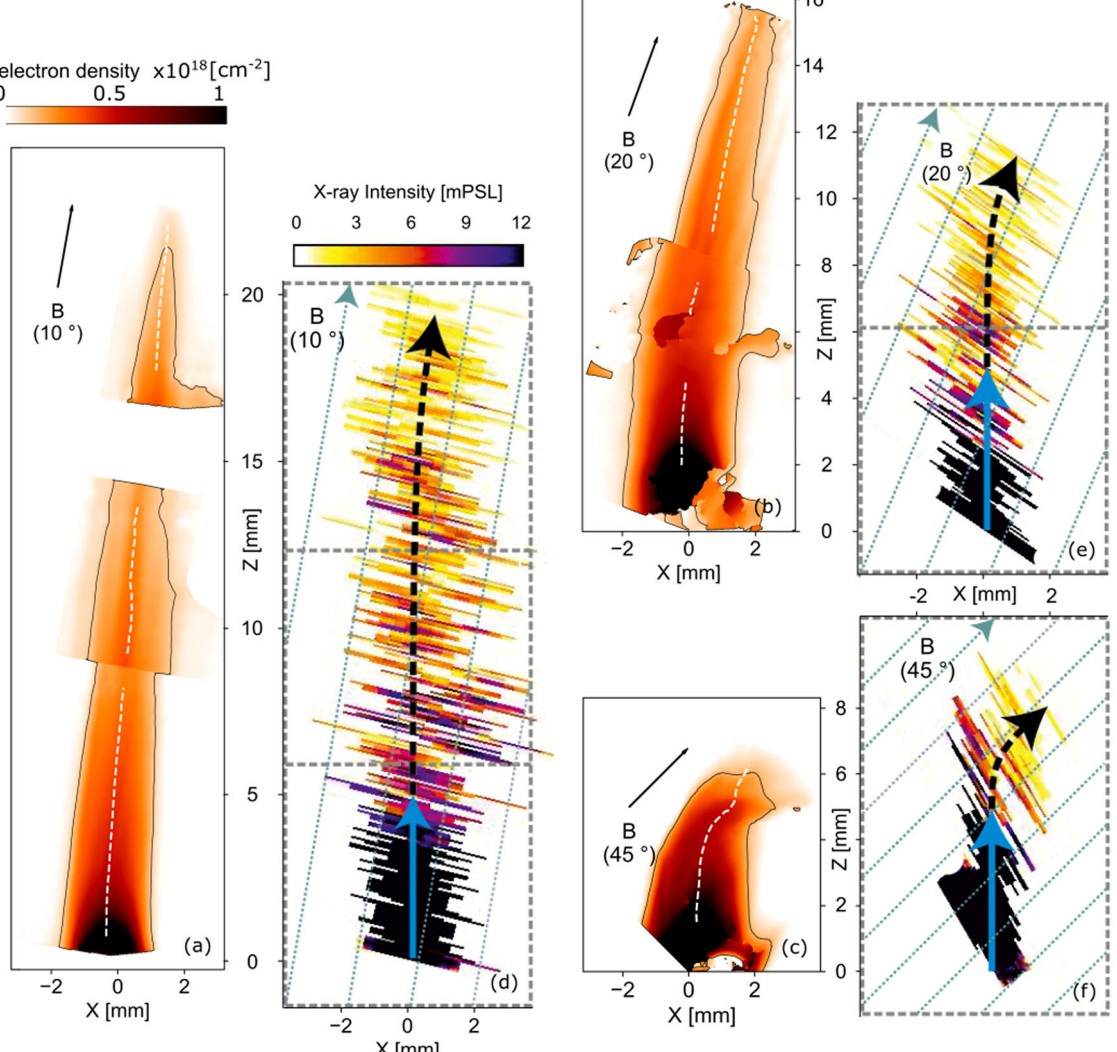

**Fig. 2 Laboratory plasma expansion observed in misaligned configurations. a** Laboratory electron density integrated along the probe line of sight, in a 20 T case at 36 ns, retrieved via interferometric measurement (see "Methods" section), with a misalignment of $\alpha = 10$ degrees between the target and the magnetic field. The target normal is along $z$ (as illustrated by the dashed gray arrows in Fig. 1A). **b** Same as **a**, with $\alpha = 20$ degrees. **c** Same as **a** and **b** with $\alpha = 45$ degrees, and at 16 ns after the laser interaction. **d, e, f** Same as **a, b, c** regarding the magnetic field angles, but as measured by X-ray spectroscopy inferred from He line emission (transition He 4p–1s on the Fluorine ion); the X-ray measurements are integrated in time (see "Methods" section and Supplementary Note 2). In **a, b, c**, the white dashed line follows the center of mass, measured from these electron density maps, assuming a constant ionization state at a given distance from the target (the lines are interrupted when the density map is noisy or when transiting from a frame to another, in order to reconstruct the full jets—see "Methods" section); the contour follows the $1 \times 10^{17} \text{cm}^{-2}$ integrated density value; the black arrows indicate the initial magnetic field direction. In **d, e, f**, the images are the result of a combination of different frames (shown by the dotted rectangles), in order to reconstruct the full jets (see "Methods" section); the dashed blue lines indicate the magnetic field direction; the solid blue and dashed black arrows are here to guide the eyes on the bending of the 2D X-rays pattern. In all cases, $z = 0$ represents the target surface.

distances from the target and to form a narrow, cylindrically shaped, extended jet (see the 45° case at 47 ns: Fig. 3). Also, there is evidence of plasma leaking, in the $xz$ plane, far from the central axis $z_{c.m.}$. This can be seen already in the 20° case (Fig. 2b, on the right-hand side of the jet from $z = 8$ mm to $z = 14$ mm), and it is even more obvious at the late time of the 45° case (Fig. 3).

Figure 4 quantifies the decreasing collimation efficiency when increasing the misalignment between the outflow and the magnetic field. There are shown experimental longitudinal lineouts of the integrated (along y) electron density measured, at 47 ns after the laser interaction, along $z_{c.m.}$ for different magnetic field angles. In the perfectly aligned case (Fig. 4a), the jet collimation induced by the magnetic field is obvious. This is witnessed by the rapid drop in electron density in the unmagnetised case (dashed lines)

that contrasts with the flatter and higher-density profile when the 20 T magnetic field is applied (solid line). Increasing the angle $\alpha$ between the magnetic field and the target normal induces a decrease of the amount of plasma along the redirection axis. Figure 4f summarizes this effect by showing the ratio $\int_y n_{e,B} dy / \int_y n_{e,noB} dy$ as a function of the magnetic field angle, taken at $z_{c.m.} = 5$ mm. Starting from the perfectly aligned case, the ratio decreases progressively to finally tend to the value of 1. It means that in this case the expansion tends to be similar to the unmagnetised one. From this experimental evidence, we infer that asymmetric plasma expansion caused by the misaligned magnetic field disturbs the formation of the cavity responsible for jet collimation[26–28], and thus prevent the efficient forming of a dense jet.

3D MHD simulations allow us to get insight into the dynamic of the jet formation or disruption for varying outflow/magnetic field angles. In Fig. 5, we show simulated maps of the integrated electron density along two lines of sight: perpendicular (a–c), i.e., as in the laboratory measurements of Figs. 1B, 2 and 3, and parallel (d–f) to the magnetic field direction. The simulations are performed using the 3D resistive MHD code GORGON (see "Methods" section).

The simulated electron density maps of Fig. 5a–c clearly corroborate the experimental maps in terms of flow redirection,

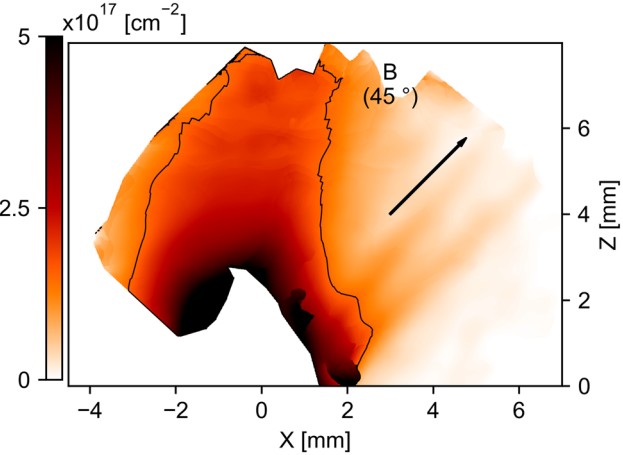

**Fig. 3 Experimental plasma expansion for large misalignment and late time.** The electron density, integrated along the probe line of sight, is shown at 45 degrees of misalignment and at 47 ns after the start of plasma expansion. The contour follows the $1.5 \times 10^{17} \text{cm}^{-2}$ integrated density value. The black arrow indicates the initial magnetic field direction.

collimation and regarding flow leakage away from the central plasma structure. In addition, the perpendicular line of sight shown by Fig. 5d–f sheds light on a strong asymmetry between the two planes (containing the magnetic field lines, $xz$ plane, and perpendicular to the magnetic field lines, $yz$) for large misalignments. The strong asymmetry for the large misalignment comes as the radial plasma expansion in the $x-$direction is increasingly more parallel to the magnetic field lines (and so free to develop), while on the other hand the plasma expansion in the $y-$direction stays perpendicular to the magnetic field lines (and so hampered).

In order to understand more in detail the lack of flow collimation as the misalignment is increased, we show in Fig. 6, superimposed to a $x$–$z$ slice of electron density, streamlines of the velocity field for the $\alpha = 10$ degrees and $\alpha = 45$ degrees cases. The generation of oblique shocks on the cavity walls is the first significant component of the global magnetic collimation process. These are fast shocks across which the flow is efficiently redirected toward the tip of the cavity, along the cavity wall. The second important component of the collimation is the formation of a diamond shock as a result of the converging flows (see Fig. 6a). These secondary fast shocks redirect the converging flows which then propagate in a direction almost parallel to the original magnetic field.

As one can see in Fig. 6a, i.e., in the $\alpha = 10$ degrees case, the flow redirection is almost perfectly symmetric on each side of the cavity. At $\alpha = 45$ degrees however (Fig. 6b) the configuration is largely asymmetric, disturbing the above mentioned schema. Indeed, while the right-side flow seems to be relatively well redirected, the left-side flow shows very little collimation as the flow encounters the magnetic field much more frontally on this side. Consequently, as the angle is increased, the flow convergence toward the cavity tip is progressively lost and the diamond shock can not effectively form. Instead, the plasma is spread in the $x$–direction, resulting in an expansion of the plasma

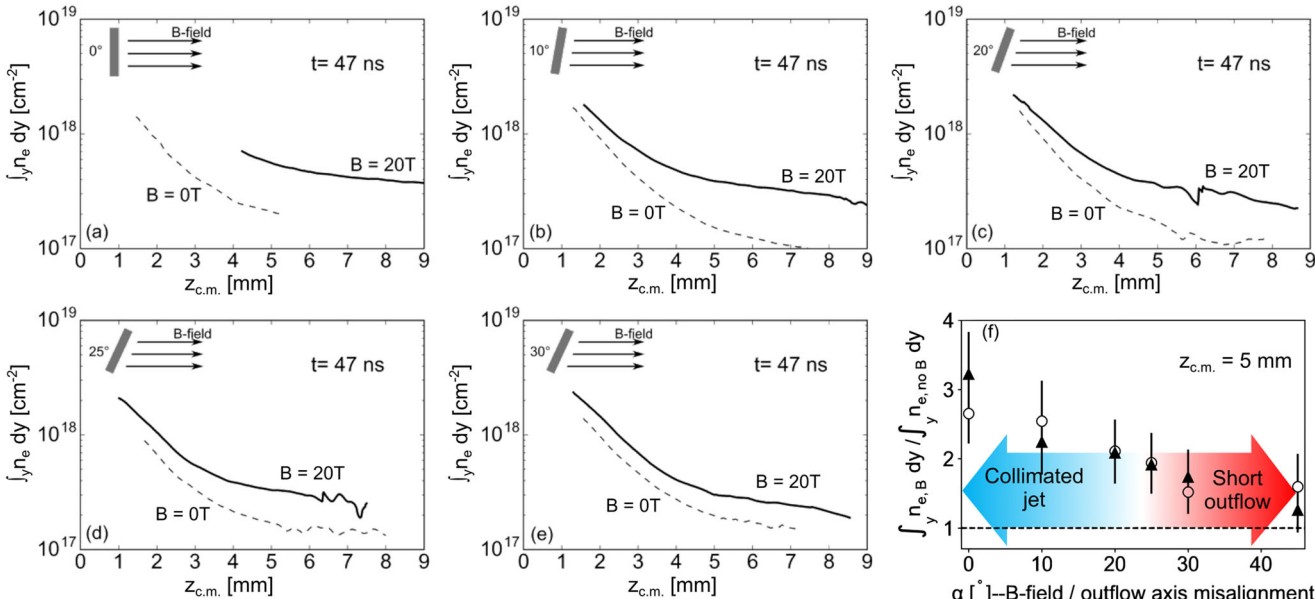

**Fig. 4 Evolution of the center of mass electron density vs. magnetic misalignment: evidence for lack of collimation.** Line of sight integrated electron density at the location of the center of mass, $\int_y n_e dy(z_{c.m.})[\text{cm}^{-2}]$ (see text; with $z_{c.m.} = 0$ being the target surface) for different target angles (going from **a** to **e**, from 0 to 30 degrees, respectively) relative to the magnetic field direction, and at a time of 47 ns after the start of the plasma expansion. The solid lines show lineouts of plasma expansion in the presence of the magnetic field, while the dashed lines are lineouts without magnetic field. The profiles are voluntarily stopped for small and high z. In the first case, the high electron density is inaccessible to optical probing, or fringes quality is too poor. In the second case, the plasma flow is out of the accessible field of view. **f** Ratio $\int_y n_{e,B} dy / \int_y n_{e,noB} dy$ as a function of the magnetic field angle vs. the target normal, taken at $z_{c.m.} = 5$ mm (which is the last point at which we measure the unmagnetized outflow in all cases), and at $t = 47$ ns. For details on the difference between the full triangles and the empty circles and about the error bars, see Supplementary Note 3.

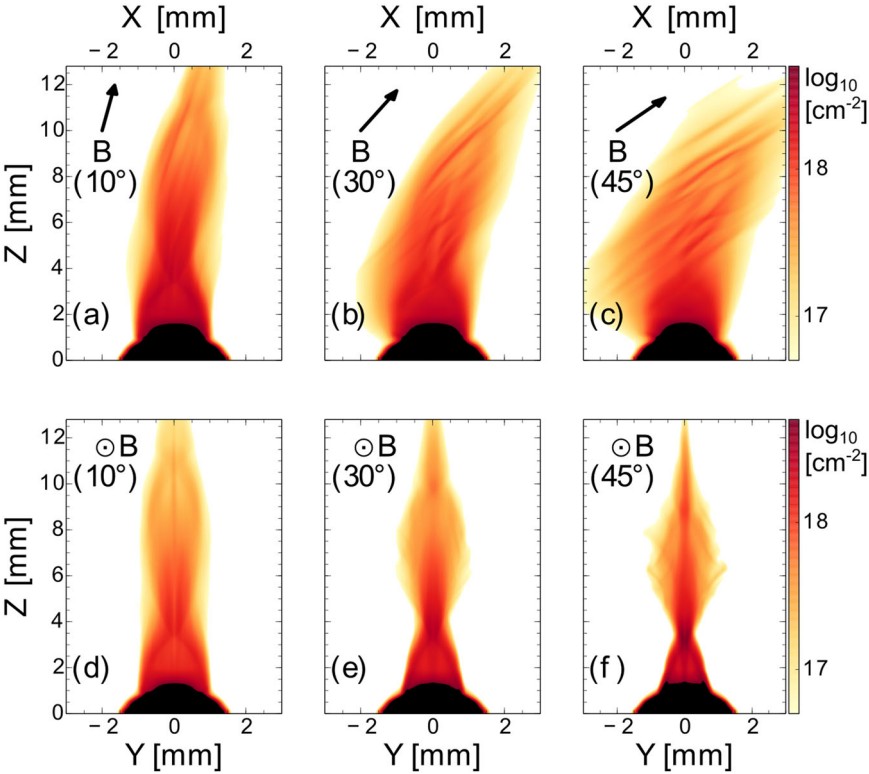

**Fig. 5 Simulated plasma expansion vs. magnetic misalignment.** 2D maps (in the $xz$ and $yz$ planes) of 3D MHD numerical simulations displaying the decimal logarithm of the integrated electron density along either $x$ or $y$. Figures **a**, **b**, and **c** correspond to maps in the plane containing the magnetic field whereas figures **d**, **e**, and **f** correspond to maps in the plane orthogonal to the magnetic field. Thus, **a**, **b**, and **c** correspond to the laboratory measurements shown in Fig. 1B, and Figs. 2 and 3. All figures correspond to a time of **18 ns** after the start of the plasma expansion. The black arrows indicate the initial magnetic field direction. The target material is made of Carbon—see "Methods" section to get insights about the simulations and the reason for the difference with the experimental target material.

perpendicularly to the magnetic field direction, pushing against the magnetic field lines more easily in the $z$–direction as the magnetic tension is reduced by that spread in the $x$–direction (see Supplementary Note 4 for more details).

## Discussion
Our results show that important effects on the plasma flow are induced by a misalignment between the outflow and a large scale, poloidal magnetic field. Among those effects are a reduced collimation due to the disruption of a symmetric collimating-cavity formation, instabilities as Rayleigh-Taylor (inducing additional leakage of matter), as well as additional heating. We stress that those precise mechanisms, while affecting specific location of the outflow, can induce specific plasma structures having important impact on the whole shape and structuring of the outflow. Those structures may be present in a wide array of astrophysical objects, and we have shown in particular a good scalability of our experimental plasmas with YSO's jets, as well as Sun's outflows. Those results then suggest that the large-scale, poloidal magnetic field (and precisely its alignment with the outflow) is an important parameter to look at when discussing the collimation and stability of outflows of matter exiting certain astrophysical systems. Indeed, as shown by our results, the mechanism responsible for the efficient collimation of a wide-angle outflow by a poloidal component of the magnetic field[26–28] is progressively disturbed for increasing misalignment angle $\alpha$ between the main plasma momentum and the magnetic field. Increasing $\alpha$ induces intrinsic asymmetry which makes the convergence at the tip of the collimating cavity increasingly inefficient. Increasing $\alpha$ will also favor the growth of the Rayleigh–Taylor instability[38], thus decreasing

even more the amount of matter distributed in a jet-like structure (see striations, witnessing Rayleigh–Taylor modes, in density maps of Figs. 3 and 5c). In Fig. 3, we can estimate that around 24% of the mass is leaking out of the central column (by integrating, in Fig. 3, the density inside and outside of the contour and considering a constant ionization in order to determine the mass).

Overall, these findings are well consistent with the observation-backed simulations of solar coronal outflows interacting with local magnetic field[1]. Our findings are also consistent with the reported observations in YSOs[36] and[37] that show a tendency for well collimated, long range, bright jets to be aligned with the magnetic field, while oppositely, weaker and shorter jets are mainly found to be misaligned with the magnetic field. As a consequence, we claim that in a situation where a well collimated outflow is observed, there is indeed good alignment between the outflow and the magnetic field.

Beyond the case of solar outflow and of YSOs, to which the laboratory plasma is directly scaled, our findings also suggest that geometrical changes in the ambient magnetic field might as well explain the variations in other astrophysical objects.

We will first discuss the direction of jets of particles escaping from bow-shock PWNe (e.g., the s-shaped jet of the Lighthouse nebula[12]) and the distortions of the tail at certain distances from the pulsar (as in the case of the Mushroom nebula[15]). Usually a strong density gradient in the interstellar medium (ISM) is invoked to explain such modifications of the tail, even if most of these sources are estimated to be in under-dense regions (with $\lesssim 0.1$ particles cm$^{-3}$). Let us consider in detail the characteristics of the mushroom nebula. It shows a bright X-ray head and a

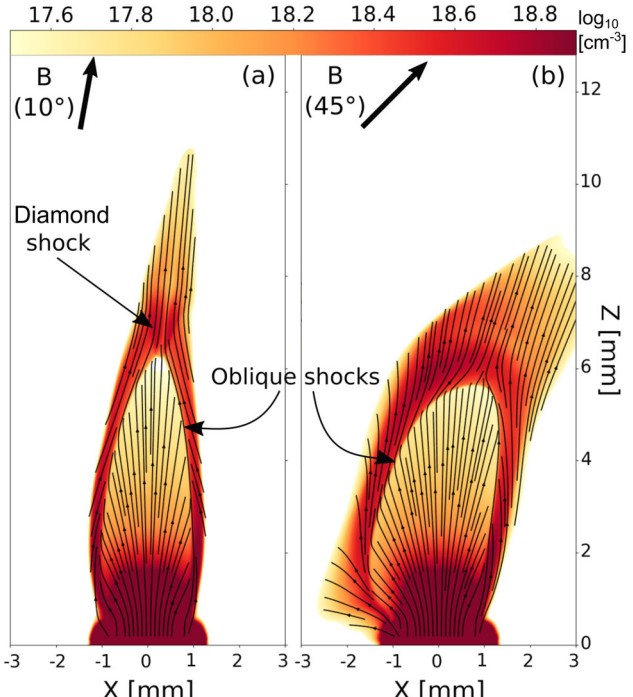

**Fig. 6 Simulation insights in the cavity formation disruption.** 2D slices of the decimal logarithm of the electron density (in the $xz$ plane) from 3D MHD GORGON simulations for **a** $\alpha = 10$ degrees and **b** $\alpha = 45$ degrees. Also shown are some velocity streamlines (black lines) displaying the differences in flow collimation as the angle is increased. Both images correspond to a time of **10 ns** after the start of plasma expansion. The target material is made of Carbon—see "Methods" section to get insights about the simulations and the reason for the difference with the experimental target material.

fainter elongated tail, extending for $L_{tail} \sim 7'$. Two faint asymmetric jets, called whiskers, extend from the head for $\sim 1.5'$ (the west one) and $\sim 3.5'$ (the east one) in a direction roughly orthogonal to the tail. These can be interpreted as formed by particles escaping from the bow shock and then tracing the structure of the underlying magnetic field. The condition for the tail to feel possible variations of the ISM magnetic field is the equipartition with the magnetic outer pressure. Considering an ambient magnetic field of order of ~5 μG, compatible with the formation of the whiskers, this condition is realized at a distance of ~$0.3 - 0.5 L_{tail}$, which roughly corresponds to the location at which the tail is seen to bend in the whiskers direction. Considering the structure of the ISM magnetic field to remain almost unchanged along the tail, the modification of the tail direction could thus be easily explained as the effect of the interaction of the tail plasma with the orthogonal magnetic field of the ISM, similarly to what is seen in our experiment. The distance at which the effect of the bending appears more evident in the laboratory framework, ~10 mm with 1 mm being the spatial scale of the experimental system (see Fig. 2 and Fig. 5), is also compatible with the one estimated for the bow shock tail, of order of $10d_0$ in the case of the Mushroom nebula, with $d_0 = \left[\dot{E}/(4\pi c \rho_{ISM} v_{PSR}^2)\right]^{1/2}$ being the spatial scale of the system (the so called stand-off distance; with $\dot{E}$ the energy losses rate of the pulsar, or luminosity; $\rho_{ISM}$ the interstellar medium density; $v_{PSR}$ the pulsar velocity).

Another interesting case to discuss in the light of the present laboratory observations is the bending of highly collimated flows which has been observed in the context of extragalactic jets. Observations reveal, for instance in the case of BL Lacerate

objects, that one-sided jet structures at parsec and kpc scales are strongly misaligned[39]. Prominent curvature effects are also detected in the peculiar morphology of wide-angle tail radio galaxies which cannot be accounted for by external forces (e.g., thermal pressure gradients). For these reasons, jet bending has been an active subject of debate in the extragalactic jet community for more than two decades, see e.g., ref. [40] and the review by ref. [41]. Although recent numerical investigations are supporting the idea of fluid instabilities-induced bending, it remains plausible that deflection could also be accounted for by the interaction with an oblique extragalactic magnetic field, as highlighted by our laboratory results. This possibility has been explored, with the aid of three-dimensional numerical simulations[42,43]. For weakly supersonic jets (or in equipartition), effective bending is observed by a relative angle of 45°. The effect is, however, suppressed for Mach number $\gtrsim 4$. Overall, the results obtained from these simulations (in the classical regime) favorably compare to the conclusions drawn in this work: the bending scale depends on the relative angle between the jet and the field, the jet velocity and the plasma magnetization. Nevertheless, relativistic MHD computations indicate that relativistic jets are less affected by the field obliquity and that bending is not as strong as in the classical case.

Finally, we note that in the laboratory configuration investigated here, the fixed angle between the outflow and the large-scale magnetic field is an idealized situation. For instance, in real YSO systems, this angle can change during the lifetime of the star due to various effects, e.g., perturbations in case of binary/multiple systems, or effects of precession or nutation in the stellar rotation axis. In this frame, we could expect the produced jet, instead of being uniform, to be highly modulated and structured. Hence, our results could provide an alternative explanation to the structuring observed in jets, in the form of chains of knots and bow structures[3,6,44,45], as due to variations in the angle between outflow and large-scale field. We note that this scenario, in which a dynamically changing magnetic field could modulate a jet, could be complementary to other scenarios involving the interaction of the outflow with an ambient medium and the proper motion of the shocks/knots resulting from that interaction[46–48]; these effects are not mutually exclusive, and could reinforce the structuring of jets. In this case, the jet structure would reflect these changes in angle and the analysis of observations of jets could provide a diagnostic to obtain information about the astrophysical system, namely the changes in the alignment of outflow and magnetic field. Similarly, observations of a s-shaped morphology of protostellar jets at parsec scales[7] could be interpreted, in the light of our results, as originating from a regular and gradual change in the direction of the ambient field, and not necessarily as due to an intrinsic precession of the jet. Indeed, if the disk axis (and the jet) is characterized by a precession but the ambient field does not change its direction, the precession should be not visible because the outflow would be always redirected in the direction of the field. However, if the jet has no precession and the ambient field changes direction, the latter would imprint a gradual change of direction of the jet on parsec scales. In short, the internal structure of YSO jets (e.g., knots, shocks) could be interpreted as the signature of changes over time in the angle between the outflow and the ambient field, when the change in the direction of the jet (as in the case of s-shaped morphology of jets) could simply be the signature of changes in the direction of the ambient field, irrespectively of the behavior of the jet.

## Methods

**Setup of the laboratory experiment.** The laboratory experiment was performed at the Elfie laser facility (LULI, Ecole Polytechnique) using a chirped laser beam of 0.6 ns duration and 40 J energy, at the wavelength of 1057 nm and focused down to a ~700 μm diameter spot on a Teflon ($CF_2$) target. Such a target was chosen so that

we could perform X-ray emission spectroscopy from the emitting F ions (see below). This gives a maximum intensity on target of $I_{max} = 1.6 \times 10^{13}$ W cm$^{-2}$.

The expanding hot Teflon plasma is coupled to an external magnetic field with large temporal ($\mu$s) and spatial (cm) scales compared to the scales of the observed laboratory plasma dynamic (100 ns, mm). The strength of the magnetic field is set to B = 20 T, via the coupling of a 32 kJ/16 kV capacitor bank delivering 20 kA to a Helmholtz coil, which is specifically designed to work in a vacuum chamber environment[28,49].

The electron density is inferred using a visible optical beam within a Mach–Zehnder interferometer arrangement. The probe beam is a 100 mJ/5 ps/ 1057 nm (1$\omega$) beam and crosses the interaction region perpendicularly to the plane in which the target is tilted, i.e., along the y-axis in Fig. 1A. This probe pulse is frequency doubled and both frequencies (1$\omega$ and 2$\omega$), co-propagating after the frequency doubling crystal, are split in two orthogonal polarizations, S and P. This arrangement yields a set of four pulses (2$\omega$-P/2$\omega$-S/1$\omega$-P/1$\omega$-S) that are temporally separated with delay lines by ~10 ns between each other. This technique allows us probing the plasma electron density at four different times for each laser shot[28]. Because of the limited field of view through the magnetic field coil (11 mm), we captured the full plasma (i.e., over several cm) evolution by moving the target, over a series of different shots, along the laser axis (z) and within the magnet assembly. This is done only over the maximum length over which the magnetic field shows little variation (we use as a criterion that B does not vary by more than 10%) which corresponds to a total length of ≈25 mm within the coil. The images thus obtained are then patched in order to get the full spatial evolution of the plasma, as shown in Figs. 1a and 2a, b. Such patching is possible due the high reproducibility of the plasma dynamics, which is due to the high reproducibility of the applied magnetic field, as it is generated by a pulse-power machine. To verify such high reproducibility, on top of the continuity observed when moving the target along the z–axis, two shots were taken in each setting and location. The reproducibility of the plasma dynamics is also attested by the similarity between the results presented here for a jet co-aligned with the magnetic field to the results we obtained in the same configuration, but in other experiments[27,28].

The plasma X-ray emission is retrieved using a a focusing spectrometer (FSSR) with high spectral and spatial resolution (about 80 $\mu$m in this experiment)[50] (see Supplementary Note 2 for details on the analysis). This spectrometer was implemented to measure the X-ray spectra emitted by the multi-charged ions from the plasma, in the range 13–16 Å (800−950 eV) in the $m = 1$ order of reflection. The spectrometer was equipped with a spherically bent mica crystal with parameters 2d = 19.9376 Å and curvature R = 150 mm. The time-integrated spectra were registered on Fujifilm Image Plate TR[51], which were placed in a cassette holder protected from optical radiation. For this, the aperture of the cassette was covered by two layers of filters made of polypropylene (1 $\mu$m) and aluminum (200 nm).

The spectrometer was aligned to record the spectrally resolved X-ray emission of the plasma with 2D spatial resolution, though with a strong astigmatism and far larger magnification factor along the jet axis than in the transverse (spectral dispersion) direction. Knowing the scaling factors in both directions, and accounting for optical distortion of the imaging system we can reconstruct a real 2D image of the jets. In order to fully reconstruct the long plasma expansion, we used different shots for which the target was located at different positions within the coil, as for the above mentioned optical measurements. Such images are shown in Fig. 2d, e, f, for various values of $\alpha$.

**Numerical simulations.** The numerical simulations were performed using the 3D Eulerian, radiative (optically thin approximation), resistive Magneto–Hydro–Dynamic (MHD) code GORGON[22,52], with the possibility to rotate the angle of the uniform 20 T magnetic field with respect to the target surface plane. The initial laser deposition (up to 1 ns on a Carbon target) is modeled in axisymmetric, cylindrical geometry with the two-dimensional, three-temperature, radiative (diffusion approximation), Lagrangian, hydro-dynamic code DUED[53], which is then passed to GORGON. The purpose of this hand-off is to take advantage of the capability of the Lagrangian code to achieve very high resolution in modeling the laser-target interaction.

Note that the Teflon target (as used for the experiment) could not be simulated in the GORGON code which cannot, in its present state, treat multi-species. In consequence, the closest choice was made, i.e., to use a carbon target instead. The purpose of the GORGON simulation is to give qualitative insight in the plasma dynamic regarding the specific interaction between a hot and highly conductive plasma and an external magnetic field (magnetic field lines being expelled out, super-Alfvenic shocks being created, plasma flow being redirected, plasma instabilities growing, etc.). The fact that the GORGON simulations are able to match well the dynamics of the experimental plasma regarding those MHD mechanisms, regardless of that difference in target material, can be seen in past studies comparing GORGON simulations with laser-driven experiments[27,28,38].

**Scalability between the laboratory and astrophysical outflows.** Looking at the plasma parameters (detailed in Table 1), we note the good scalability of this laboratory dynamics to solar coronal outflow[1], as well YSO jets (e.g., the DG Tau A object and its associated HH 158 jet[54,55]). We stress that we consider here the scalability to YSO outflows within the region 10 to 50 AU from the star, i.e., to

outflows having already encountered a pre-collimation from about 90° to 25° full opening angle in the near zero to 10 AU region. Indeed, it can be gathered from astrophysical observations that outflows emerge with quite large initial divergence (up to 86° full opening angle[55]) from a launching region that is few AU wide in radius, centered on the star[55,56]. Then they are progressively collimated within a distance of about 50 AU[55–57] leading, in the case where a narrow jet forms, to small divergence angles (only few degrees), which are compatible with the expected radial expansion of a supersonic collimated jet[58,59]. We also note that this setup has already shown consistency between laboratory observations and astrophysical ones, namely a steady diamond shock at the top of the cavity corroborating steady X-ray emissions in YSO jets (as discussed in refs. [27,48,60]). The scalability is ensured by all plasmas being well described by ideal MHD. This is attested by the fact that the relevant dimensionless parameters, namely the Reynolds number ($R_e = L \times v_{stream}/\nu$; $L$ the characteristic size of the system; $v_{stream}$ the flow velocity; $\nu$ the kinematic viscosity[61]), Peclet number ($P_e = L \times v_{stream}/\chi_{th}$; $\chi_{th}$ the thermal diffusivity[61]) and Magnetic Reynolds number ($R_m = L \times v_{stream}/\chi_m$; $\chi_m$ the magnetic diffusivity[62]) are, in the three cases considered in Table 1, much greater than the unity. Indeed, although these numbers can differ by orders of magnitude, the fact that they are much greater than the unity ensures that the momentum, heat and magnetic diffusion are negligible with respect to the advective transport of these quantities. We also verify that the Euler ($Eu = v\sqrt{\rho/P}$) and Alfvén ($Al = B/\sqrt{\mu_0 P}$) numbers are close enough between the laboratory and natural systems in order for them to evolve similarly[61,63].

Once the scalability between the laboratory system and a particular natural system (here, that of a YSO jet or that of a solar outflow) are assured, through the matching between their Euler and Alfvén numbers, the quantitative correspondence between the laboratory and the natural parameters can be done as follows[62]. We use the fact that, both in the laboratory and natural cases, the spatial scale, the velocity and the density are known (see Table 1). From this, it is possible[62] to establish the following relations for the magnetic field and temporal evolution of both systems : $B_{astro} = c \times \sqrt{b} \times B_{labo}$ and $t_{labo} = (a/c) \times t_{astro}$, where $a = (r_{astro}/r_{labo})$; $b = (\rho_{astro}/\rho_{labo})$; $c = (v_{astro}/v_{labo})$. From these relations, we can state that 10 ns of laboratory evolution corresponds to 4 months of astrophysical evolution of a YSO jet, and to 55 s of a solar outflow. Similarly, we can state that the $2 \times 10^5$ G (20 T) magnetic field in the laboratory corresponds to 29 mG for the YSO jet magnetic field, and to 2 G for a solar outflow. Note that the value of the YSO jet magnetic field, retrieved this way via the scaling relations, is very consistent with the inferred value of the ambient magnetic field: 24 mG (as detailed in Supplementary Note 1)—that last value is the one quoted in Table 1. In the case of the solar outflow, the magnetic filed value retrieved via the scaling relations matches well the magnetic field observed at the top of a magnetic loop (~3 G).

As anticipated in section "Results", a general scaling to the case of PWNe, in analogy to what done for the other objects, is almost pointless. The first motivation is that different systems may have quite different spatial scales, since the stand-off distance $d_0$ depends on the properties of the ambient medium (density) and the pulsar (velocity and luminosity), so that a general scale cannot be quantified; the scaling must then refer to a single object. On the other side many of the parameters given in Table 1 for YSO and solar outflows can be only barely constrained from observation in case of (some) PWNe, such as the velocity of the flow in the tail (far from the wind injection zone), the magnetic field and density of the ISM, the particles density in the tail or in the orthogonal jets. This makes the attempt of giving a general recipe for scaling between the laboratory setup to the case of PWNe not much significant, while a qualitative comparison, as described in the main text, can be done based on energetic arguments.

## Data availability
The data that support the findings of this study are available from the corresponding authors upon reasonable request.

## Code availability
The code used to generate Figs. 5 and 6 is GORGON, using data from DUED. Both codes are detailed in the Methods section.

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

## Acknowledgements

We thank the LULI teams for technical support, the Dresden High Magnetic Field Laboratory at Helmholtz–Zentrum Dresden–Rossendorf for the development of the pulsed power generator, B. Albertazzi and M. Nakatsutsumi for their prior work in laying the groundwork for the experimental platform, and P. Loiseau (CEA) for discussions. This work was supported by the European Research Council (ERC) under the European Union's Horizon 2020 research and innovation program (Grant Agreement No. 787539). M.S. was supported by Russian Foundation for Basic Research, project No. 18-29-21018. The JIHT RAS team's work is supported by the grant No. 13.1902.21.0035 in the form of a subsidy from the Ministry of Science and Higher Education of the Russian Federation. This work was partly done within the LABEX Plas@Par project and supported by Grant No. 11-IDEX-0004-02 from ANR (France). The reported study was funded by Russian Foundation for Basic Research, project No. 19-32-60008. O.W. would like to acknowledge the DFG Programmes GRK 1203 and SFB/TR18. All data needed to evaluate the conclusions in the paper are present in the paper. Experimental data and simulations are, respectively, archived on servers at LULI and LERMA laboratories and can be consulted upon request. Part of the experimental system is covered by a patent (n° 1000183285, 2013, INPI-France). This work was granted access to the HPC resources of MesoPSL financed by the Region Ile de France. The research leading to these results is supported by Extreme Light Infrastructure Nuclear Physics (ELI-NP) Phase II, a project co-financed by the Romanian Government and European Union through the European Regional Development Fund, and by the project ELI_RO_2020_23 funded by IFA (Romania). We acknowledge the use of JSCC RAS computational resources. This work was performed under the auspices of the U.S. Department of Energy by Lawrence Livermore National Laboratory (LLNL) under Contract DE-AC52-07NA27344.

## Author contributions

A.C., D.P.H., and J.F. conceived the project. G.R., E.F., J.B., M.C., S.N.C., T.G., D.P.H., M.O., M.I.S., S.P., and J.F. performed the experiments, with support from M.S. and O.W. G.R., E.F., S.N.R., I.Yu.S., S.P., and J.F. analyzed the data. T.V. performed and analyzed the DUED simulations. B.K. and A.C. performed and analyzed the GORGON simulations. G.R., B.K., A.C., and J.F. wrote the bulk of the paper, with major contributions from E.F., C.A., R.B., A.M., B.O., S.P., and S.O. All authors commented and revised the paper.

## Competing interests

The authors declare no competing interests.
