## [Peer Review File · Nature Communications]

Editorial Note: This manuscript has been previously reviewed at another journal that is not operating a transparent peer review scheme. This document only contains reviewer comments and rebuttal letters for versions considered at Nature Communications

Reviewers' Comments:

Reviewer #2:

Remarks to the Author:

We appreciate the implementation of our comments into the current manuscript and the detailed answers proposed to address our main concerns.

We appreciate the addition of the sun's coronal outflow application that expands the public to which this type of comparison is addressed.

We think this paper is suited for publication in Nature Communications.

Reviewer #3:

Remarks to the Author:

Summary of key results:

The paper details laboratory experiments & simulations which demonstrate that magnetic collimation of pressure-driven flows becomes weaker with greater flow-field misalignment.

Originality & significance

These are the first jet experiments using an oblique poloidal magnetic field and can be dimensionlessly scaled to several astrophysical contexts (YSOs, solar flows, PWes, etc.). Lab astrophysics experiments such as these are critical to understanding these objects and represent a means of verifying simulations and testing models.

Data and Methods

The significant mismatches between simulation and experiment (Carbon vs. Teflon, etc.) should be more clearly explained and discussed earlier in the paper.

General Comments

The paper is much improved and resolved most of the concerns from the initial natureastro submission. However, the current text of the paper overreaches occasionally in attempting to explain the astrophysical significance. For example, the first line of the discussion, "Our results suggest that the large-scale collimation of the outflow in astrophysical systems depends mostly on the orientation between the ejection axis and the ambient magnetic field". This statement omits or contradicts many of the qualifications (e.g. lack of collimation from toroidal fields, unknown scaling for PWNes, scaling

limited to specific regions of different objects) stated earlier.

It is suggested that the significance could be more powerfully rephrased in terms of "scalable structures/mechanisms/effects" rather than fully scalable analogs to astrophysical systems. -> Experimental measurements & sims demonstrate that misaligned field + flows create skewed shock structures. These skewed shock structures induce several scalable effects: reduced collimation, additional heating, more instabilities (RT, leakage)... Skewed shock structures may be present in a wide array of astrophysical objects including... and the experiments dimensionlessly scale to YSO jets, solar flows... consequently, this work has broad relevance and significance.

This may resolve many of the previous reviewer concerns about direct analogs/relevance and still explain the significance of the experiments.

mechanisms, regardless of that difference in target material, can be seen in past studies comparing GORGON simulations with laser-driven experiments [Khar, B. *et al.* (2019) 'Laser-produced magnetic-Rayleigh-Taylor unstable plasma slabs in a 20 T magnetic field', *Physical Review Letters*, 123(20), pp. 205001–205001. doi: [10.1103/PhysRevLett.123.205001](https://doi.org/10.1103/PhysRevLett.123.205001). ; Higginson, D. P. *et al.* (2017) 'Enhancement of Quasistationary Shocks and Heating via Temporal Staging in a Magnetized Laser-Plasma Jet', *Physical Review Letters*, 119(25), pp. 255002–255002. doi: [10.1103/PhysRevLett.119.255002](https://doi.org/10.1103/PhysRevLett.119.255002). ; Albertazzi, B. *et al.* (2014) 'Laboratory formation of a scaled protostellar jet by coaligned poloidal magnetic field.', *Science*, 346(6207), pp. 325–8. doi: [10.1126/science.1259694](https://doi.org/10.1126/science.1259694).]

General Comments:

The paper is much improved and resolved most of the concerns from the initial natureastro submission. However, the current text of the paper overreaches occasionally in attempting to explain the astrophysical significance. For example, the first line of the discussion, "Our results suggest that the large-scale collimation of the outflow in astrophysical systems depends mostly on the orientation between the ejection axis and the ambient magnetic field". This statement omits or contradicts many of the qualifications (e.g. lack of collimation from toroidal fields, unknown scaling for PWNes, scaling limited to specific regions of different objects) stated earlier.

It is suggested that the significance could be more powerfully rephrased in terms of "scalable structures/mechanisms/effects" rather than fully scalable analogs to astrophysical systems. -> Experimental measurements & sims demonstrate that misaligned field + flows create skewed shock structures. These skewed shock structures induce several scalable effects: reduced collimation, additional heating, more instabilities (RT, leakage)... Skewed shock structures may be present in a wide array of astrophysical objects including... and the experiments dimensionlessly scale to YSO jets, solar flows... consequently, this work has broad relevance and significance.

We thank the reviewer for this comment about the global relevance of our results, and which gives us the opportunity to state, in a better and more precise way, the real significance of those results. Taking into consideration the reviewer's latest comment, we have then rephrased the first sentence of the *Discussion* section (as pointed out by the reviewer) as follows:

Our results show that important effects on the plasma flow are induced by a misalignment between the outflow and a large scale, poloidal magnetic field. Among those effects are a reduced collimation due to the disruption of a symmetric collimating-cavity formation, instabilities as Rayleigh-Taylor (inducing additional leakage of matter), as well as additional heating. We stress that those precise mechanisms, while affecting specific location of the outflow, can induce specific plasma structures having important impact on the whole shape and structuring of the outflow. Those structures may be present in a wide array of astrophysical objects, and we have shown in particular a good scalability of our experimental plasmas with YSO's jets as well as Sun's outflows. Those results then suggest that the large-scale, poloidal magnetic field (and precisely its

alignment with the outflow) is an important parameter to look at when discussing the collimation and stability of outflows of matter exiting certain astrophysical systems.

Indeed, as shown by our results, the mechanism responsible for ...

Additional change:

As already mentioned previously to the reviewers (following their review made for Nat. Astro.), the value of the magnetic field in the Table 1, for the YSO, had to be changed from 25 mG to 24 mG (due to a rounding error). This was not done at the time of the last submission. This is now made.

Reviewers' Comments:

Reviewer #3:

Remarks to the Author:

No further comments. Recommend for publication.